# Lipid Nanoparticles as a Shuttle for Anti-Adipogenic miRNAs to Human Adipocytes

**DOI:** 10.3390/pharmaceutics15071983

**Published:** 2023-07-19

**Authors:** Anna-Laurence Schachner-Nedherer, Julia Fuchs, Ivan Vidakovic, Oliver Höller, Gebhard Schratter, Gunter Almer, Eleonore Fröhlich, Andreas Zimmer, Martin Wabitsch, Karin Kornmueller, Ruth Prassl

**Affiliations:** 1Gottfried Schatz Research Center for Cell Signaling, Metabolism and Aging, Division of Medical Physics and Biophysics, Medical University of Graz, 8010 Graz, Austria; anna.schachner-nedherer@medunigraz.at (A.-L.S.-N.); julia.fuchs@medunigraz.at (J.F.); ivan.vidakovic@medunigraz.at (I.V.); oliver.hoeller@medunigraz.at (O.H.); karin.kornmueller@medunigraz.at (K.K.); 2Department of Pharmaceutical Technology and Biopharmacy, Institute of Pharmaceutical Sciences, University of Graz, 8010 Graz, Austria; andreas.zimmer@uni-graz.at; 3Clinical Institute for Medical and Chemical Laboratory Diagnostics, Medical University of Graz, 8010 Graz, Austria; gunter.almer@medunigraz.at; 4Center for Medical Research, Medical University of Graz, 8010 Graz, Austria; eleonore.froehlich@medunigraz.at; 5Division of Pediatric Endocrinology, Diabetes Department of Pediatrics and Adolescent Medicine, University Medical Center Ulm, 89075 Ulm, Germany; martin.wabitsch@uniklinik-ulm.de

**Keywords:** microRNA, lipid nanoparticles, adipogenesis, automated quantitative image analysis

## Abstract

Obesity and type 2 diabetes are major health burdens for which no effective therapy is available today. One treatment strategy could be to balance the metabolic functions of adipose tissue by regulating gene expressions using miRNAs. Here, we have loaded two anti-adipogenic miRNAs (miR26a and miR27a) into a pegylated lipid nanoparticle (PEG-LNP) formulation by a single-step microfluidic-assisted synthesis step. For the miRNA-loaded LNPs, the following system properties were determined: particle size, zeta potential, miRNA complexation efficiency, and cytotoxicity. We have used a human preadipocyte cell line to address the transfection efficiency and biological effects of the miRNA candidates at the gene and protein level. Our findings revealed that the upregulation of miR27a in preadipocytes inhibits adipogenesis by the downregulation of PPARγ and the reduction of lipid droplet formation. In contrast, miR26a transfection in adipocytes induced white adipocyte browning detected as the upregulation of uncoupling protein 1 (UCP1) as a marker of non-shivering thermogenesis. We conclude that the selective delivery of miRNAs by PEG-LNPs to adipocytes could offer new perspectives for the treatment of obesity and related metabolic diseases.

## 1. Introduction

Adipocytes were recognized early as regulators of energy balance [1]. Dysfunctions in adipocyte metabolism and lipid storage are, therefore, closely linked to the progression of chronic diseases including obesity or type 2 diabetes (T2D), which are amongst the major health burdens worldwide [2,3]. To manage the metabolic functions of adipose tissue, targeted modifications in fat metabolism might offer promising strategies for therapeutic interventions and pharmacological treatment of obesity and other metabolic diseases [4]. During adipogenesis, adipocyte precursor cells develop into mature adipocytes, which accumulate to form adipose tissue. Basically, adipose tissue can be broadly divided into two major classes with different functions: white adipose tissue (WAT) is mainly responsible for fat storage containing large lipid droplets; in contrast, brown adipose tissue (BAT) fulfills energy-dissipating functions, containing multiple small lipid droplets and densely packed mitochondria [5]. Both types of fat are important for the regulation of the overall energy balance in humans [6,7]. A third category of adipocytes are brown-in-white, brite, or beige adipocytes [8,9]. Brite adipocytes are formed upon cold exposure or induced by agonists as either long-term β-adrenergic receptor or peroxisome proliferator-activated receptor γ (PPARγ) activation [10]. In vitro, browning can be induced during differentiation from preadipocytes [11] or by the trans-differentiation of mature white adipocytes [12,13,14]. Brite adipocytes have a high capacity for glucose and lipid uptake. Thus, the successful biogenesis of thermogenic brown-like (brite) adipocytes, which burn fat into heat and equally reduce plasma levels of glucose and fatty acids, is a promising alternative concept for the treatment of obesity [15,16,17] or T2D [18]. Accordingly, stimulating BAT formation and the energy-dissipating function of adipocytes might present a novel approach for anti-adipogenic therapy.

Even though numerous studies have identified adipose tissue as potential therapeutic targets in obesity [18,19,20], no therapeutic drug that specifically targets adipose tissue is clinically approved until now [19,21]. In this regard, microRNAs (miRNAs) have emerged as a novel class of regulatory determinants in adipogenesis and energy homeostasis [22]. Today, numerous miRNAs have been identified in human adipose tissue and a large number of studies have demonstrated that miRNAs play an important role in adipocyte differentiation, lipid storage, glucose homeostasis, and insulin resistance [23,24]. While some miRNAs activate brown adipogenesis, others are brown adipogenesis inhibitors [22,25]. Other miRNAs directly impact adipocyte lipid composition and lipid droplet formation by either the direct or indirect targeting of transcription factors [26]. MiRNAs are short non-coding RNAs of 19 to 25 nucleotides in length, which regulate gene expression by translational repression, exerting multiple biological functions. Among the group of anti-adipogenic miRNAs, miR26a and miR27a are two representative candidates, which could be considered for therapeutic approaches.

MiR26a/b were the first miRNAs identified to induce and promote brite adipocyte differentiation in a human adipocyte cell line, namely, human multipotent adipose-derived stem (hMADS) cells [27]. Both miRNAs of the miR26a family differ only in two nucleotides sharing the same seed sequence [28]. The inhibition of both miR26a family members significantly reduced lipid accumulation and the expression of characteristic adipocyte markers in mature adipose tissue including bioactive adipokines like adiponectin or leptin [27]. In contrast, increasing the expression of miR26a/b by the transfection of mimics modestly accelerated adipogenesis, causing browning. Brite thermogenic adipocytes have more mitochondria than WAT, resulting in a strong upregulation of uncoupling protein 1 (UCP1) that is substantially abundant in mitochondria [9]. UCP1 has the function to uncouple mitochondrial respiration from adenosine triphosphate (ATP) synthesis, which creates heat and energy dissipation [29]. Mechanistically, the positive effects of miR26a/b on the browning of WAT are largely mediated through their direct repression of ADAM metallopeptidase 17 (ADAM17), a gene that was previously described to negatively regulate non-shivering thermogenesis [30,31].

MiR27a/b were originally identified as negative regulators of adipogenesis in hMADS, being functionally linked to white adipogenesis [32,33]. Both anti-adipogenic miRNAs share the same seed sequence differing in only one nucleotide [34]. In this context, they exhibit a strong anti-adipogenic function due to the direct targeting and repression of PPARγ, which is considered as the master regulator of adipogenesis and adipocyte differentiation [35,36]. Beyond that, different profiles of miRNA expression in preadipocytes and mature adipocytes point to the role of miRNAs in the regulation of adipogenic differentiation [37]. There is strong evidence that miR27a inhibits adipogenic differentiation as shown in 3T3-L1 preadipocytes, by the significant reduction of intracellular fat accumulation and the inhibition of adipocyte formation [38]. This finding was further confirmed using mouse-derived multipotent mesenchymal stem cells [39]. Overall, these results imply that members of the miR27 family might act as general adipogenic inhibitors and regulators of adipocyte expansion during obesity.

Although miRNAs can be applied in cell culture without any specific delivery system, commercially available transfection reagents of little-known chemical composition are routinely used to promote miRNA’s cellular uptake. Different transfection reagents may also show different efficiencies and off-target effects [40]. Moreover, such transfection reagents are not well-suited for in vivo experiments in preclinical settings. It thus became increasingly important to design proper delivery systems for miRNA- and RNA-based therapeutics in general. A range of viral and non-viral systems have been explored so far, with cationic liposomes and lipid nanoparticles (LNPs) being widely studied [41]. The latter predominantly contain ionizable cationic lipids, which are neutrally charged at physiological pH, but become positively charged in the acidified endosome after cellular uptake [42]. Additionally, LNP formulations are often enriched with a pegylated lipid to form a polymer shell surrounding the LNPs with the aim to improve particle stability and enhance biological half-life.

In the last few years, numerous oligonucleotide-based therapeutics entered clinical trials; among them are only a few miRNA-based therapeutics. Moreover, some of the studies with miRNAs had to be discontinued due to adverse side effects in patients [43,44]. However, to date, four small interfering (siRNA)-based products have been approved for clinical use in humans [45]. The first siRNA therapeutics, named patisiran, which was approved for the treatment of hereditary transthyretin amyloidosis, is an LNP formulation marketed as Onpattro, comparable to the LNP-mRNA vaccines recently launched for COVID-19 [46]. The basic lipid composition of Onpattro is an ionizable lipid, a pegylated lipid, a neutral lipid, and cholesterol [47]. In this study, we have used the same lipid composition and lipid ratios as reported for Onpattro. Here, the LNPs were synthesized using a continuous flow microfluidics device. We have selected two representative miRNAs, namely, miR26a and miR27a, as oligonucleotide drug candidates that have previously been reported to show anti-adipogenic activity. For the pegylated LNPs (PEG-LNPs), the following system properties were characterized: particle size, zeta potential, miRNA complexation efficiency, and cytotoxicity. Further, we have used the human Simpson Golabi Behmel Syndrome (SGBS) preadipocyte cell line to address the transfection efficiency and biological effects of the miRNA candidates at a gene and protein level. Our results show that miR26a/27a can be efficiently complexed with a clinically approved LNP formulation to show specific regulatory effects in human adipocytes. Our data demonstrate that miRNA-complexed PEG-LNPs could be considered as a promising delivery system, specifically addressing metabolic effects in adipogenesis.

## 2. Materials and Methods

The ionizable cationic lipid (6Z,9Z,28Z,31Z)-heptatriaconta-6,9,28,31- tetraen-19-yl-4-(dimethylamino) butanoate (DLin-MC3-DMA; CAS Number. 1224606-06-7) was obtained from THP Medical Products (Vienna, Austria). The polyethylene glycol functionalized lipid (R)-2,3- bis(tetradecyloxy)propyl-1-(methoxy polyethylene glycol 2000) carbamate (PEG-DMG; CAS Number 1397695-86-1), 1,2-distearoyl-sn-glycero-3-phosphocholine (DSPC; CAS Number 816-94-4), and cholesterol (CAS Number 57-88-5) were purchased from Merck (Darmstadt, Germany). HiPerFect Transfection Reagent (Cat. No./ID: 301704) was obtained from Qiagen (Hilden, Germany). MiR26a (hsa-miR26a-5p UUCAAGUAAUCCAGGAUAGGCU), miR27a (hsa-miR27a-3p UUCACAGUGGCUAAGUUCCGC), non-targeting control (NTC) based on cel-miR-67 UCACAACCUCCUAGAAAGAGUAGA, and fluorescent-labelled NTC-miRNA mimic transfection control with Dy547 (miRNA-CY3) comprising the sequence CUCUUUCUAGGAGGUUGUGA 5´ Dy547 were obtained from THP Medical Products (Vienna, Austria). Nuclease micrococcal from Staphylococcus aureus was acquired from Merck (Vienna, Austria). RediPlate ^TM^ 96 RiboGreen RNA Quantification Kit (R-32700) was obtained from Molecular Probes (Thermo Fischer Sci., Vienna, Austria).

### 2.1. Preparation of miRNA-Complexed Pegylated Lipid Nanoparticles (PEG-LNPs)

Amino lipid (D-LinMC3-DMA; 64.7 µg), helper lipid (DSPC; 15.9 µg), cholesterol (30.0 µg), and DMG-PEG2000 (7.6 µg) were dissolved in 500 µL of ethanol absolute at a molar ratio of D-LinMC3-DMA:DSPC:cholesterol:DMP-PEG2000 of 50:10:38.5:1.5. Nucleic acids (miR26a, miR27a, NTC) exhibiting a stock concentration of 20 µM dissolved in 50 mM citrate buffer (pH 4.0); 40 µL of the miRNA stock solution were diluted to a volume of 1500 µL with citrate buffer. The miRNA/amino lipid charge ratio corresponded to a nitrogen-to-phosphate (N/P) ratio of 3. For microfluid-based assembling, the lipid-containing ethanolic and nucleic-acid-containing aqueous phases were mixed in a herringbone geometric microchannel device (NanoAssemblr^®^, Precision NanoSystems, Vancouver, BC, Canada) using a glass cartridge. A flow rate of 2 mL/min and flow ratio of 3:1 corresponding to 1500 µL of aqueous and 500 µL of organic phase were used. Final concentrations of nucleic acids and lipids in the PEG-LNP complex were 400 nM and 59.1 µg/mL, respectively. For control experiments, empty PEG-LNPs were prepared in a similar way by mixing 10 mM citrate buffer (pH 4.0) with lipids. To remove the ethanol, the suspensions were subsequently dialyzed by using the Float-A-Lyzer^®^ G2 Dialysis Device (MWCO 100 kDa) against 10 mM PBS buffer solutions (150 mM NaCl; pH 7.5 or pH 5.5) overnight.

The commercially available transfection reagent HiPerFect (QIAGEN, Hilden, Germany) was used for comparison and as positive control. The complexation of nucleic acids (miR26a, miR27a, NTC) with HiPerFect representing a mixture of cationic and neutral lipids was performed according to the manufacturer’s instructions. In brief, a working solution of nucleic acids (4 µM in 10 mM citrate puffer pH 4.0) was mixed with the HiPerFect lipid mixture in a volumetric ratio of 1:1.6 (nucleic acids:HiPerFect), incubated for 10 min at room temperature (about 23 °C), and further diluted with DMEM/F12. The final concentration of nucleic acids in the HiPerFect complexes was 377 nM. For control experiments, empty particles were prepared by diluting HiPerFect with RNase-free water.

The miRNA quantification is performed using RediPlate^TM^ 96 RiboGreen RNA Quantification Kit (R-32700). The kit contains preloaded RiboGreen reagent that binds to the RNA, resulting in a bright green fluorescence signal. miRNA standards and 10 mM Tris-HCl, 1 mM EDTA buffer pH (7.5) were used to create a standard curve in a concentration range of 5–1000 ng/mL. In the first sample set, 20 µL of each PEG-LNP (directly after preparation and after dialysis) were diluted with above-mentioned buffer to the final volume of 200 µL. In another set of samples, 10 µL of 10% Triton X-100 were added to cause particle disruption and miRNA release. All dilution and mixing steps were performed directly in the 96-well plate containing the RiboGreen reagent. After 10 min incubation, the fluorescence intensity was recorded at Ex/Em of 480-10/520-15 nm using CLARIOstar plate reader (BMG LABTECH GmbH, Ortenberg, Germany). The experiments were run in duplicate.

### 2.2. Particle Size and Zeta Potential Measurements

By using dynamic light scattering (DLS) and electrophoretic light scattering (ELS) technologies, particle sizes (intensity-based size distribution, Z-average) and zeta potentials were determined for all nanoparticle samples at 25 °C, utilizing a Zetasizer Nano ZS from Malvern Instruments (Malvern Panalytical, Malvern, UK). The device setup is equipped with a green laser (532 nm). Particle size measurements were carried out using UVette cuvettes (Eppendorf, Hamburg, Germany) with back-scattering detection at 173 °C. For analysis, the data-processing model was set to general. The polydispersity index (PDI) was estimated using the PDI implemented in the Malvern evaluation software (Malvern Panalytical, Malvern, UK).

Before measurements, the samples were diluted 1:10 (*v*/*v*) with an aqueous sodium chloride solution (0.9% (*w*/*v*) NaCl; conductivity 50 µS/cm). For ELS measurements, the PEG-LNP samples were titrated in a pH range from 5.5–7.5 by adjusting the pH value with 0.1 M HCl. Clear folded capillary cells from Malvern Instruments were used. For DLS and ELS experiments, all measurements were performed in triplicate.

### 2.3. Complexation Capacity

To assess the extent of complex formation between nucleic acids and lipids, 1.5–2% agarose gels were prepared using 1 µL of GelRed in 1× Tris-Acetate-EDTA (TAE) buffer. All nanoparticle samples (PEG-LNP, HiPerFect) and the controls, including dsRNA Ladder (New England Biolabs, N0363S), as well as pure nucleic acids, were applied in comparable concentrations to the gel. Finally, the samples were treated with 1% Triton X-100 to release the miRNA. In some cases, endonuclease was added to induce miRNA digestion.

The gel was placed into an electrophoresis tank (Embi Tec, RunOne Elecrophoresis cell; Thermo Fisher Scientific, Vienna, Austria) and covered with 1× TAE buffer. The power supply voltage was set to 75 V. After running for a duration of 35–40 min, the gel was transferred onto the UV-tray and scanned with BioRad GelDoc EZ Imager (BioRad Laboratories, Hercules, CA, USA). Finally, data processing and analysis were performed using the Image Lab Software (BioRad, version 6.1).

### 2.4. Cell Culture Experiments

Human Simpson Golabi Behmel Syndrome (SGBS) cells were kindly provided from the research group of Dr. Wabitsch (University Medical Center Ulm, Germany).The preadipocytes were derived from the stromal cell fraction of subcutaneous adipose tissue of an infant with SGBS, a rare genetic condition that affects many parts of the body and occurs primarily in males [48]. Optimized protocol for SGBS fat cell differentiation is described in detail by Fischer-Posovszky et al. [49]. In brief, the morphologically fibroblast-like preadipocytes were cultivated at 37 °C under 5% CO_2_ water-saturated atmosphere in complete proliferation medium (PM) consisting of DMEM/F12 (1:1), supplemented with 0.9% biotin-pantothenate-mixture (final concentrations of 33 µM and 17 µM of biotin and pantothenate in DMEM/F12 (1:1), respectively), 0.2% normocin, and 10% FBS. To extracellularly trigger the in vitro differentiation of SGBS preadipocytes at confluency of ~90% (time point is stated as induction at day 0 = d0), freshly prepared induction medium (IM) was added, consisting of serum-free PM, containing final concentrations of 20 nM human insulin, 0.01 mg/mL apo-transferrin, 2 µM rosiglitazone, 0.2 nM triiodothyronine, 0.2 nM hydrocortisone, 0.25 mM IBMX, and 25 nM dexamethasone. The medium was replaced every four days (d4, d8, and d12) during the differentiation period depending on the specific timing of miRNA transfection and the corresponding differentiation regime. After a differentiation period of four days (d4), the composition of the differentiation medium (DM) was slightly adopted using serum-free PM containing final concentrations of 20 nM human insulin, 0.01 mg/mL apo-transferrin, 0.2 nM triiodothyronine, and 0.2 nM hydrocortisone.

Transient transfection with nucleic acid complexes was performed in either non-confluent SGBS preadipocytes (d-1 for miR27a) or SGBS adipocytes (d8 for miR26a). In addition to the metabolically active miR26a and miR27a, NTC was used as a negative control. After an incubation period of 24 h, the transfection medium was changed to either IM (d0 for miR27a) or DM (d9 for miR26a). PBS washing was performed to remove unbound transfection sample material. To obtain final nucleic acid concentrations of 20 nM and/or 40 nM, in vitro transfection was performed by pipetting the respective volumes of nucleic acid complexes on the cells.

According to the specific miRNA effects on adipogenesis, different endpoints and evaluation methods were determined as described in the following sections.

### 2.5. RNA Isolation and Reverse Transcription

To determine mRNA levels of PPARγ (after miR27a treatment) and UCP1 (after miR26a treatment), SGBS cells were seeded on 12-well plates (Greiner Bio-One GmbH, Frickenhausen, Germany). In vitro experimental conditions were performed in duplicate. RNA harvest of adipocytes was performed at d5–d6 (for miR27a) and at d14–d16 (for miR26a).

Total RNA was isolated using the Fatty Tissue RNA Purification Kit (Norgen Biotek Corperation, ON, Canada) according to manufacturer’s protocol. Sample concentrations and purity (absorbance ratio A260/A280) were measured with NanoDrop^®^ ND-1000 (Peqlab, Avantor, Radnor, PA, USA). Reverse transcription was performed using the High-Capacity cDNA Reverse Transcription Kit (Thermo Fisher Scientific, Waltham, MA, USA). According to the manufacturer’s protocol, RT Puffer (10×), 0.8 µL of dNTP (100 mM), 2 µL of RT Random Primer (10×), nuclease free water, and MultiScribe Reverse Transcriptase (1 µL) were added to the purified RNA (maximum of 2 µg total RNA). The thermocycler program was set to 10 min at 25 °C, 120 min at 37 °C, 5 min at 85 °C, and, subsequent, cooling at 4 °C. The resulting cDNA was diluted (1:50 in RNase-free water) and stored at −80 °C.

### 2.6. Real-Time Quantitative PCR (RT-qPCR)

RT-qPCR was performed in 96-well plates (BioRad, Hercules, CA, USA) with cDNA concentrations between 25–30 ng/µL using QuantiTect SYBR^®^ Green PCR kit (Cat.no. 204143; Qiagen, Hilden, Germany) according to the manufacturer’s protocol. Detailed information on primer sequences is provided in Appendix A. Assays were run on a C1000TM Thermal Cycler (BioRad, CFX96 Real-Time System, or CFX384 Real-Time System), choosing the following cycle times: 15 min at 95 °C, 39 cycles of 10 s at 95 °C, and 1 min at 60 °C, with holding temperature of 12 °C. All samples were measured in triplicate. Data were analyzed with Bio-Rad CFX Manager 3.1 software. After melting curve analysis, the relative mRNA expression levels of PPARγ and UCP1, normalized to the housekeeping gene GAPDH, were calculated by using the 2ΔΔCt value.

A two-tailed Student’s *t*-test, assuming unequal variances, was used to estimate the statistical significance of differences between means using GraphPad PRISM software (version 6.01). Results were considered statistically significant when *p* < 0.05 (* *p* < 0.05, ** *p* < 0.01, *** *p* < 0.001, and **** *p* < 0.0001).

### 2.7. Oil Red O (ORO) Staining

For Oil Red O staining, SGBS preadipocytes were seeded as duplicates on chamber slides and transfected on d1 with either miR27a or NTC complexed with PEG-LNPs or HiPerFect with a final nucleic acid concentration of 40 nM. As a control, non-transfected preadipocytes and adipocytes were used. After 24 h, the process of differentiation into adipocytes (d0) was initiated, followed by a media change at d4. At d6, chamber slides were washed with PBS, fixed in 3.7% paraformaldehyde for 10 min, followed by a washing step with 60% (*v*/*v*) 2-propanol. A stock solution of ORO was prepared by dissolving 0.35 g dye in 100 mL of 2-propanol. ORO working solution was prepared by mixing the stock solution in a volumetric ratio of 3:2 with distilled water, followed by filtration. After 15 min incubation with ORO working solution, chamber slides were rinsed with 60% 2-propanol. Nuclei were counterstained using hematoxylin and slides were mounted with Kaiser’s glycerol gelatin (Merk Millipore, Darmstadt, Germany).

### 2.8. Immunofluorescence Staining

To perform fluorescence staining, mature SGBS adipocytes on chamber slides (seeded as duplicates) were transfected (d8) with either miR26a or NTC complexed with PEG-LNPs or HiPerFect with a final nucleic acid concentration of 40 nM. Non-transfected preadipocytes and adipocytes served as controls. Medium change was performed on d9 and d12. A multicolor fluorescent staining was performed at d14 to identify the mitochondrial carrier protein UCP1 as well as mitochondrial structures and actin filaments. MitoTracker™ Orange stock solution (1 mM; MitoTracker™ Orange CM-H2TMRos; Thermo Fisher Scientific; Cat. No.: M7511) was diluted to 250 nM using Dulbecco’s Modified Eagle Medium/Nutrient Mixture F-12 (Gibco Cell Medium; Thermo Fisher, Vienna, Austria) supplemented with 1% L-glutamine (200 mM; Thermo Fisher, Vienna, Austria) and incubated on the living SGBS culture for 15 min at 37 °C and 5% CO_2_. Chamber slides were washed with 1xPBS and fixed in 3.7% paraformaldehyde for 10 min. Subsequently, cells were permeabilized for 10 min with 0.2% Triton X-100 solution and blocked for 15 min at RT with blocking buffer solution (0.2% Triton X-100 and 2% BSA in PBS). The primary polyclonal rabbit anti-human UCP1 antibody (1.3 mg/mL) was diluted 1:1000 in antibody diluent and slides incubated overnight at 4 °C. Slides were washed with 1×PBS and incubated using the secondary Alexa Fluor 633-conjugated goat-anti-rabbit-antibody (1:200, Thermo Fisher Scientific, Cat. No.: A21070) for 30 min at RT. After PBS washing, slides were treated with Hoechst 33342 (1:2000) and Alexa FluorTM 488 Phalloidin (Invitrogen™, Cat. No.: A12379) for 15 min. Finally, sections were mounted with ProLong™ Gold Antifade Reagent (Thermo Fisher Scientific).

### 2.9. Image Acquisition and Quantitative Image Analysis

Brightfield as well as fluorescence images were taken using the Olympus SLIDEVIEW VS200 slide scanner (Olympus, Hamburg, Germany) equipped with LED illumination and the fluorescence light source X-Cite XYLIS (Excelitas Technologies. Waltham, MA, USA). The 20× objective (UPLXAPO 20×/0.80; Olympus) was used for capturing both the brightfield and fluorescence images. A mirror cube filter set (Pentafilter F66-987_OEF) was used for visualization of DAPI, FITC, CY3, CY5, and CY7 channels. The exposure time of the CY3 and CY5 channel was set using “Auto Exposure” function on the most intense staining of each treatment, providing the shortest possible exposure time (CY3-83,1 ms; CY5-141,998 ms). The same exposure time was used for all images throughout all treatments to allow comparison of fluorescence intensities within a group. A constant exposure time was used for the FITC and CY7 channel to detect background fluorescence. Scan overviews were adapted and exported using OlyVIA software (version 3.4.1, Olympus).

For miR27a experiments, brightfield images of ORO staining were acquired by scanning a total area of 50 mm^2^ (~20,000 cells) per treatment. For quantitative image analysis of ORO staining, each scan per treatment was divided into 200 single tiles using ZEN 3 blue software (Version 3.0; Carl Zeiss). The open-source image analysis software CellProfiler (version 3.1.8; McQuin et al. [50],) was used to measure the lipid area (red staining) and the number of nuclei per image for each treatment condition. In brief, the original brightfield images were split into HSV channels using the “Color to grey” module. To identify lipid droplet areas, saturation channel images were processed with the “Identify Primary Objects” module. To mask lipid droplet areas on original greyscale images, the module “MaskImage” was used. Subsequently, the “ImageMath” module was used with the operation tool “invert”. A thresholding with a lower bound of 0.12 and an upper bound of 1 was applied with the threshold strategy “Global” and the thresholding method “Robust Background” using the module “Threshold”. The parameter “Threshold correction factor” was set to 1 while the “Threshold smoothing scale” was set to 30. The module “Identify Primary Objects” was used to determine the number of nuclei in each image. Using the “Overlay Objects” module, we created overlay images of lipid droplet areas or cell nuclei. The “Measure Area Occupied” tool was used to calculate the lipid droplet area per image, that was exported to an Excel Sheet with the “ExportToSpreadsheet” tool. For each treatment, lipid areas were normalized to nuclei number. A schematic illustration of automated quantitative image analysis of ORO staining is provided in Appendix A. The statistical significance of differences between means was assessed using a one-way ANOVA with Tukey’s multiple comparison test. Results were considered statistically significant when *p* < 0.05 (* *p* < 0.05, ** *p* < 0.01, *** *p* < 0.001, and **** *p* < 0.0001).

For miR26a experiments addressing quantitative image analysis of UCP1 and mitochondria levels, images of multifluorescent staining were evaluated by scanning a total area of 20.5 mm^2^ (8000–9000 cells) per treatment of the channels DAPI, FITC, CY3, CY5, and CY7. The scan of each treatment and channel was divided into 200 single tiles using ZEN 3 blue software (Version 3.0; Carl Zeiss). For each treatment condition, the mean UCP1 level intensity (red, CY5 channel), the mean mitochondrial level intensity (yellow, CY3 channel), and the number of nuclei (blue, DAPI channel) were measured using the cell image analysis software CellProfiler (version 3.1.5; [50]). In brief, DAPI images were processed with the “Identify Primary Objects” module to determine the number of cell nuclei. Unspecific signals occurring in the CY3 (mitochondria) and CY5 (UCP1) channel were filtered by subtracting the CY7 channel (no target) from the CY5 as well as the CY3 channel using the module “ImageMath”. Following subtraction, the “IdentifyPrimaryObjects” module was used to identify the objects found in the CY3 and CY5 channels. Signal intensities of UCP1 and mitochondrial objects were then determined using the tool “MeasureObjectIntensity”. Using the “ExportToSpreadsheet” tool, the counted number of nuclei for each image and the calculated mean intensities per image were exported to an Excel Sheet. The intensity of CY3 and CY5 objects was normalized to the number of nuclei per image. Mean fluorescence intensities of the CY3 and CY5 signals within a treatment group were compared using one-way ANOVA with Tukey’s multiple comparison test. Results were considered statistically significant when *p* < 0.05 (* *p* < 0.05, ** *p* < 0.01, *** *p* < 0.001, and **** *p* < 0.0001).

### 2.10. Confocal Laser Scanning Microscopy (cLSM)

For cLSM experiments, SGBS cells were seeded onto glass-bottom dishes commonly used for fluorescence imaging (WillCo Wells B.V., Amsterdam, The Netherlands). To trace miRNA uptake after transfection, fluorescent-labeled miRNA-CY3 was complexed with PEG-LNPs or HiPerFect at a final nucleic acid concentration of 40 nM. cLSM images were taken at different time points (4 h and 24 h) after incubation at 37 °C. Sample conditions for preadipocytes and adipocytes were chosen according to the SGBS cell cultivation and transfection protocol as described before.

The cLSM (LSM 510 META Axiovert 200 M Zeiss confocal system, Carl Zeiss Meditec AG, Jena, Germany), in combination with the ZEN2009 software package, was used to record fluorescence images. Alexa FluorTM 488 Phalloidin was excited at 488 nm and detected using a bandpass filter (BP 505/550 nm). The nuclear marker Hoechst 33342 (Thermo Fisher Scientific) was excited at 405 nm and detected using a bandpass filter (BP 420/480 nm). The fluorescent-labelled miRNA-CY3 was detected at 543 nm excitation wavelength using a long pass filter (LP 560 nm).

### 2.11. Cell Viability

Human Simpson Golabi Behmel Syndrome (SGBS) preadipocytes were seeded in 96-well plates and either cultured as preadipocytes (~90% confluence) or differentiated into adipocytes up to d8. Preadipocytes and adipocytes were transfected and incubated for 24 h with PEG-LNPs or HiPerFect containing either miR26a, miR27a, or NTC at final nucleic acid concentrations of 40 nM (representing the experimental in vitro conditions used in this study) or 80 nM (representing deliberately induced stress conditions). Non-miRNA-loaded (empty) particles were used as controls. Cell viability was determined by a colorimetric MTS assay, and a Cell Titer 96^®^ Aqueous Non-Radioactive Cell Proliferation Assay (Promega Corporation, Madison, WI, USA) applied according to the manufacturer’s instructions. After incubating for 2 h at 37 °C, the absorbance was measured at 490 nm using a UV-/VIS-plate reader (Fluostar Galaxy, BMG Labtech GmbH, Ortenberg, Germany). Untreated, blank-corrected wells represented 100% viability (termed as control).

### 2.12. Statistical Analysis

All statistical analyses were performed using GraphPad PRISM software (version 6.01) (GraphPad Software, La Jolla, CA, USA). Quantification and statistical analyses are given in the respective methods’ description.

## 3. Results

### 3.1. Preparation and Characterisation of miRNA-Complexed PEG-LNPs

Pegylated lipid nanoparticles (PEG-LNPs) composed primarily of ionizable lipids, phosphatidylcholine (DSPC), and cholesterol enriched with a low amount of PEGylated lipid (1.5 mol%) were produced by a single-step synthesis protocol using a microfluidic-based system. The advantages of the microfluidics technology compared to conventional batch synthesis or crossflow ethanol injection are higher batch-to-batch reproducibility, scalable laboratory production volumes, and reduced particle heterogeneity due to the precisely defined process parameter, such as flow rates and flow velocities in the microfluidic channels [51,52]. In general, the size and surface charge of the LNPs are important parameters for drug delivery in controlling LNP performance. Here, a constant miRNA/amino lipid charge ratio of 3 was used throughout all experiments. Using a constant flow ratio of 3:1 (aqueous to ethanol) for the PEG-LNP fabrication protocol, the final ethanol concentration was about 25% (*v*/*v*), which required the removal of ethanol by dialysis. We first investigated whether the pH value of the buffer (acidic or neutral) has any impact on the physicochemical characteristics and the complexation stability of PEG-LNPs.

Before dialysis, the particle size values (Z-average) for miRNA-loaded PEG-LNPs were between 88–106 nm with PDI values ranging between 0.1–0.23 (n = 3). After dialysis against PBS pH 7.5 or pH 5.5, the sizes remained almost the same. The particle size values for empty PEG-LNPs were in the same range as for miRNA-loaded particles, but the PDI values were higher (Table 1). Interestingly, upon storage for 7 days at 4 °C, the empty PEG-LNPs were largely aggregated, showing a broad size distribution with particle sizes ranging up to 1 µm and PDI values varying between 0.5–1.0. In contrast, all miRNA-loaded PEG-LNPs remained stable in solution without any remarkable change in particle size and PDI value.

For comparison, miRNA-loaded particles using a commercially available transfection reagent were produced by the incubation of an aqueous miRNA solution with HiPerFect (pH 6.4) as described above. After dilution (1:10 *v*/*v*) with PBS pH 5.5 or pH 7.5, the particle sizes were about 1500 and 400 nm with PDI values of 0.6 and 0.3, respectively. A similar size distribution and equally high PDI values were recorded for HiPerFect diluted with pure buffer (pH 5.5 and pH 7.5) without miRNA. Overall, the data indicate a more homogeneous preparation of smaller-sized PEG-LNP particles fabricated via microfluidics compared to the batch production procedure used for HiPerFect particles following the manufacturer´s standard protocol.

To determine the surface charge of the PEG-LNPs, zeta potential measurements were performed using ELS. Immediately after preparation, empty PEG-LNPs showed a slightly negative zeta potential of −2.2 mV at pH 7.5 and a slightly positive zeta potential of 0.7 mV at pH 5.5 (Figure 1A, a); accordingly, both values can be considered as neutral on average [53]. MiRNA-loaded PEG-LNPs showed a similar behavior with a slightly negative zeta potential of −2.9 mV at pH 7.5 and a slightly positive net charge of 3.6 mV at a lower pH (pH 5.5). The slightly negative values for PEG-LNPs at pH 7.5 could be due to the surface-exposed PEG-lipids and the ionizable lipid (D-LinMC3-DMA) that has a pKa value of 6.44 [54]. Thus, at a nearly physiological pH, the PEG-LNPs are almost neutral. After dialysis against PBS pH 5.5 and adjusting the pH to 7.5, the PEG-LNPs again showed a slightly negative zeta potential of about –1.8 ± 1.5 mV. Decreasing the pH to 5.5, the titration curve shows a transition from negative to positive zeta potential values (Figure 1A, b), a behavior that is expected for ionizable lipid-containing formulations [54,55]. However, when the PEG-LNPs are dialyzed against PBS pH 7.5 instead of pH 5.5, they showed negative zeta potentials varying between −10 mV to −7 mV for pH values ranging from 5.5 to 7.5 as seen in the titration curve (Figure 1A, c). Thus, we speculated that parts of the miRNA could be released during dialysis against the neutral buffer, resulting in a negative zeta potential at all pH values. This assumption was later supported by gel electrophoresis experiments and quantitative analyses of miRNA by fluorescence.

In contrast, the HiPerFect solution (pH 6.4) that contains cationic and neutral lipids showed positive zeta potential values for pH 5.5–7.5 varying between 6.1 ± 0.7 mV and 16.3 ± 0.4 mV, being highest at pH 6.0–7.0 (Figure 1A, d). Complexation of HiPerFect with miRNAs led to a shift in the zeta potential from positive to negative values of −23.7 mV and −24.7 mV at pH 7.5 and pH 5.5, respectively, with less negative values determined for pH 6.0–7.0 (Figure 1A, e).

To determine whether miRNA became efficiently complexed by PEG-LNPs, agarose gel electrophoresis was performed. For the applied N/P ratio of 3, we found that the miRNA could be quantitatively complexed during the fabrication process of PEG-LNPs by microfluidics (Figure 1B, lane 2). However, after dialysis against PBS pH 7.5, a band corresponding to free miRNA (about 40% of the total amount of miRNA) became visible in the gel (Figure 1B, lane 3). As mentioned before, it is reasonable to assume that parts of miRNA are released during dialysis and are loosely associated with the PEG chains located at the surface of the LNPs, but could not be removed by dialysis. This released miRNA could readily be digested by nuclease (Figure 1B, lane 8). Upon the addition of Triton X-100 followed by the disruption of the PEG-LNPs, miRNA is completely released, showing a highly intense spot representative of free miRNA (Figure 1B, lanes 5, 6, and 7). In contrast, PEG-LNPs dialysed against PBS pH 5.5 did not reveal free or released miRNA (Figure 1B, lane 4) and the incorporated miRNA was not accessible to nuclease digestion (Figure 1B, lane 9). The behaviour was the same for both miR26a and miR27a. Even after storage for 3 weeks at 4 °C, the PEG-LNPs remained stable without any further release of miRNA (Appendix A). The results from the agarose gel experiments perfectly correspond to the surface charge measurements. A comparable behaviour, although not as pronounced as seen in the gel electrophoresis experiments, was observed in the quantitative miRNA assay. To quantify the miRNA, the PEG-LNPs have to be disrupted by the addition of Triton X-100, resulting in a final concentration of c = 351.6 ± 38.2 µg/mL miRNA (n = 7) in all samples including the free miRNA used as reference. A direct comparison with the measured fluorescence signal intensities before treatment with Triton X-100 revealed that about 11% of the miRNA were released upon dialysis against PBS pH 7.5 while only 4% were released when the samples were dialysed against PBS pH 5.5. (n = 2). To sum up, the results of both methods (zeta potential and gel electrophoresis) supported by quantitative analyses indicate that the miRNA-complexed PEG-LNPs should rather be kept at an acidic pH until use to prevent the partial release of miRNA.

The mixing of miRNA with HiPerFect, irrespective of whether miR26a or miR27a was used, resulted in a quantitative complexation of miRNAs, which could be released from the HiPerFect/miRNA complex upon the addition of Triton X-100. The storage behaviour of HiPerFect/miRNA was similar compared to PEG-LNPs miRNA complexes (Appendix A).

### 3.2. Cellular Uptake of Anti-Adipogenic miRNAs Complexed with PEG-LNPs

To visualize the interaction and uptake behavior of miRNA-loaded PEG-LNPs by SGBS cells, PEG-LNPs were complexed with fluorescent-labelled miRNA-CY3. HiPerFect/miRNA-CY3 complexes served as positive control. The experiments were performed in preadipocytes and differentiated adipocytes. Data were recorded after 4 h and 24 h of incubation. As seen in Figure 2 (left panels), the uptake of PEG-LNPs and control HiPerFect particles in SGBS preadipocytes was comparable to each other and time-dependent. After 24 h of incubation, elevated amounts of miRNA could be visualized intracellularly. The single vertical and orthogonal z-scans (shown on the right and upper side of each image, respectively) indicate the presence of miRNA-CY3 fluorescence across different cell depths. Thus, miRNA-CY3 has been successfully transferred into the cellular cytoplasm and did not predominantly stick to the cell surface. A very similar uptake behavior for miRNA was observed for mature adipocytes (d8) being transfected with PEG-LNPs and HiPerFect/miRNA-CY3 complexes (right panels in Figure 2). Again, miRNA-CY3 was observed intracellularly; however, we could not detect an increased uptake after 24 h of incubation. It should be mentioned that we have not directly labelled the lipid nanoparticles with a fluorescence dye in our experimental setup. Thus, any analytical data on the uptake and “fate” of the PEG-LNP particles themselves are not available.

The cell viability was not impaired by incubation with any of the PEG-LNPs/miRNA complexes, not even at a twice-as-high concentration (80 nM miRNA, double concentration of lipids) as used for the transfection experiments performed in this study. The data of the MTS assay for preadipocytes and adipocytes exposed to PEG-LNPs as well as the control experiments with HiPerFect are shown in Appendix A. In none of the cases, the cell viability was decreased below 90% after an incubation time of 24 h, indicating that PEG-LNPs are very well-tolerated by preadipocytes and adipocytes independent of miRNA loading. Even without information on the uptake and downstream processing of the PEG-LNP particles themselves, the results suggest that the PEG-LNPs are not toxic in the applicated concentrations.

### 3.3. Biological Effects of Anti-Adipogenic miRNAs Complexed with PEG-LNPs

To study human adipogenesis, SGBS cells constitute a valuable in vitro model to investigate the anti-adipogenic effects of miRNAs. While miR27a is affecting energy-storing lipid accumulation in white adipocytes via the downregulation of PPARγ, which represents the master regulator of adipogenesis, miR26a is influencing the energy-dissipating properties by promoting UCP1 expression and, consequently, triggering the transformation of energy-storing white into energy-dissipating brite adipose tissues [22,27]. As the anti-adipogenic gatekeepers miR26a and miR27a influence adipogenesis at different metabolic stages, miRNA-specific transfection timings and endpoints have to be considered. MiR27a directly targets PPARγ, thereby inhibiting lipid droplet formation during adipogenesis. Therefore, transfection with miR27a has to be performed in SGBS preadipocytes (d1) to trigger inhibiting effects during adipocyte development [32,39,56]. To evaluate the impact of miR27a upregulation, the reduced intracellular accumulation of lipid droplets was assessed by ORO staining [57,58]. In contrast, miR26a has to be applied in mature adipocytes (d8) to induce adipocyte browning, which can be detected by an elevated number of mitochondria, increased intracellular levels of UCP1. and a reduction in the amount of lipid droplets.

To investigate modifications in lipid metabolism at transcriptional levels, RT-qPCR analyses were performed to study the intracellular effects of miR26a (UCP1 mRNA, Figure 3) and miR27a (PPARγ mRNA, Figure 4) complexed with PEG-LNPs at final nucleic acid concentrations of 20 nM and 40 nM. The results showed a three- to four-fold upregulation in UCP1 mRNA levels for PEG-LNP/miR26a-treated cells compared to cells transfected with PEG-LNP/NTC (Figure 3A). In contrast, the positive control of HiPerFect/miR26a-transfected cells showed at two- to three-fold upregulation in UCP1 miRNA levels compared to HiPerFect/NTC-transfected cells (Figure 3B). NTC served as the non-targeting control and, thus, did not actively intervene in metabolic processes. While UCP1 was almost undetectable in low-fat control preadipocytes, it was present in lipid-containing control mature adipocytes (at d14-16) showing comparable levels to NTC-transfected cells. This observation underlines the fact that PEG-LNPs did not affect adipocyte differentiation by themselves and were at least as sufficient in miRNA delivery and the upregulation of UCP1 as the commercial transfection reagent HiPerFect.

Next, the transcriptional levels of PPARγ mRNA were assessed after the transfection of preadipocytes with miR27a complexed with PEG-LNPs at final nucleic acid concentrations of 20 nM and 40 nM. The RT-qPCR results revealed a concentration-dependent reduction in PPARγ mRNA levels (Figure 4). For PEG-LNP/miR27a complexes, a 0.4- to 0.9-fold reduction in PPARγ mRNA levels was found compared to PEG-LNP/NTC (Figure 4A). For the positive control, HiPerFect/miR27a-transfected cells (Figure 4B) demonstrated a comparable high (0.5- to 0.9-fold) downregulation of PPARγ mRNA compared to HiPerFect/NTC-transfected cells. As expected, very low levels of PPARγ were detected in low-fat control preadipocytes while lipid-containing control mature adipocytes at d5-d6 had a comparable high level of PPARγ as cells treated with NTC-containing samples.

The anti-adipogenic effects of miR27a and miR26a described at the transcriptional level were also reflected at protein levels as highlighted in Figure 5 and Figure 6.

We developed a methodology for quantitatively analyzing and measuring the mean fluorescent signal intensities of mitochondria and UCP1 protein levels across large histological image areas. According to quantitative image analysis of fluorescent-stained images, adipocytes transfected with miR26a using PEG-LNPs showed significantly higher mitochondrial fluorescence signal intensities than cells transfected with NTC (Figure 6A,C, first rows). HiPerFect/miR26a was used as a positive control and revealed similar results in terms of mitochondrial level increase as PEG-LNPs. Transfection with PEG-LNP/NTC and HiPerFect/NTC showed no significant differences in mitochondria signal levels.

Quantitative image analysis of UCP1 fluorescence revealed results that were consistent with mitochondrial levels. High UCP1 fluorescence signal levels were observed in adipocytes transfected with miR26a using PEG-LNPs and HiPerFect, whereas low signal intensities were observed in cells transfected with NTC-complexed particles (Figure 6B,C, second rows). Overlay images of mitochondria and UCP1 staining showed a colocalization of the fluorescence signals (Figure 6C, third and fourth rows).

We examined an area of approximately 50 mm^2^ corresponding to an average cell number of 25,000 for each treatment. The large area and high number of analyzed cells highlight the fundamental benefit of automated computer-assisted analysis, which resulted in an objective interpretation of the data [59,60]. This allowed for a more accurate description of immunostaining results compared to analyzing only small image sections, as is often the case with conventional histological studies. As a result, our established automated quantitative image analysis pipeline for investigating mitochondrial and UCP1 fluorescence levels served as a powerful tool for reducing operator errors. However, qualitative assessment of fluorescence staining is always required to properly evaluate immunofluorescence specificity and to avoid data misinterpretation [59,60,61].

## 4. Discussion

Lipid nanoparticles are amongst the most clinically advanced delivery vehicles for oligonucleotides, whose functionality and transfection efficacy strongly depend on their lipid components [62]. In our study, we have selected a lipid composition that closely resembles that of the first approved and marketed LNP-siRNA drug for targeting hepatocytes in the treatment of polyneuropathy in people with hereditary transthyretin-mediated amyloidosis, a fatal rare disease [63]. In contrast to the launched product, we have used a microfluidic technique for LNP preparation [47]. Instead of siRNA, we have complexed miRNAs at a constant amino-lipid-to-miRNA charge ratio of 3.

The objective of the study was to target human adipocytes using pegylated LNP/miRNA complexes as potential future therapeutics for the treatment of metabolic diseases, in particular, obesity and diabetes. For that, known anti-adipogenic miRNAs (miR26a and miR27a) were complexed with PEG-LNPs and a human preadipocyte cell strain was used to assess human adipocyte development and metabolism in vitro [64]. Our results demonstrate that the transfection efficiency of the miRNA-complexed PEG-LNPs is comparable or even higher compared to the commercially available liposomal transfection reagent HiPerFect, which was used as a positive control in this study. Our data show that miR27a directly effects adipogenesis by modulation of the transcription factor PPARγ. Likewise, lipid droplet formation was impaired, thus modulating adipose tissue development and differentiation [8]. In contrast, the transfection of mature adipocytes with PEG-LNP/miR26a led to the browning of white adipocytes as seen by an increased number of mitochondria and higher expression levels of UCP1. These results are in accordance with previous observations showing that miR26a upregulation promotes adipocyte browning, thereby triggering fat consumption and energy expenditure [27].

Taken together, our data suggest that the established PEG-LNP delivery system for siRNA [47] could be adopted as a universal delivery vehicle for miRNAs as well. Moreover, the PEG-LNP system seems to be well-suited for targeting adipocytes and transcriptional regulators in adipogenesis. Therefore, this lipid-based shuttle for anti-adipogenic miRNAs represents a novel therapeutic approach in the treatment of metabolic diseases with a focus on obesity and diabetes.

## Figures and Tables

**Figure 1 pharmaceutics-15-01983-f001:**
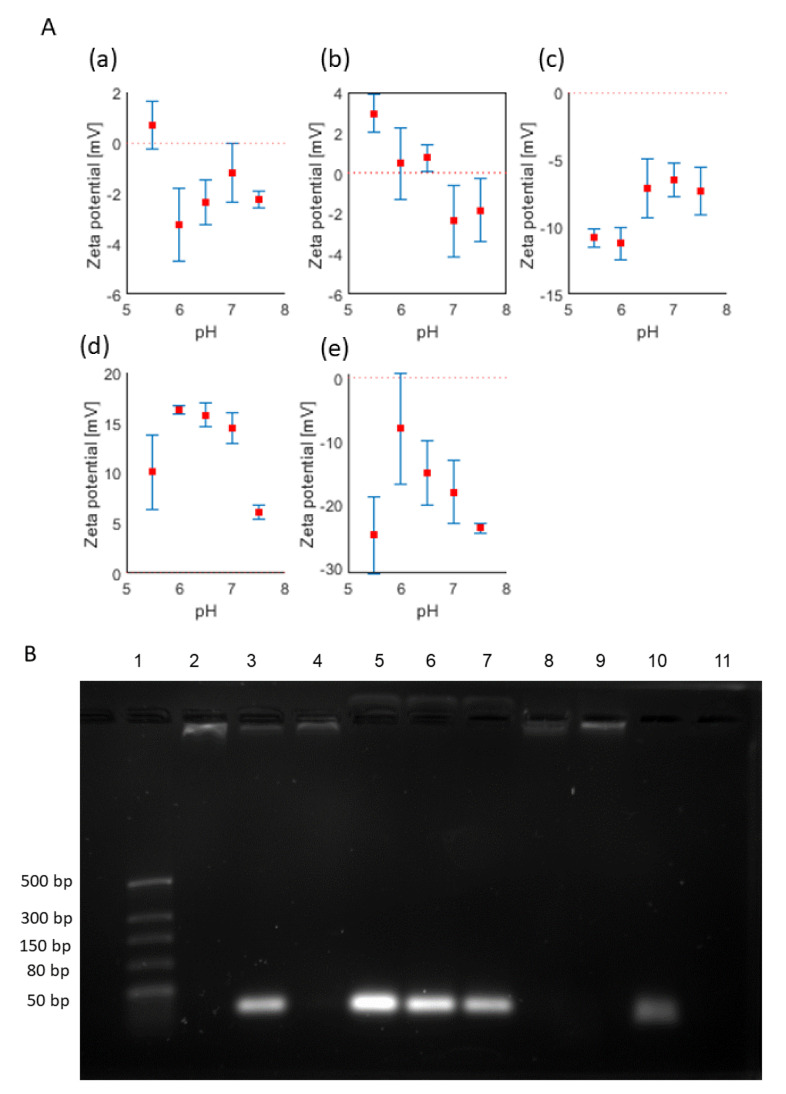
(**A**) Zeta potential titration curves for PEG-LNPs. (**a**) Empty PEG-LNPs showed a positive net charge at low pH (pH 5.5) and a transition to negative values when the pH was increased. (**b**) The same behaviour was observed for PEG-LNP/miRNA formulations when the samples were dialysed against PBS, pH 5.5. (**c**) When the samples were dialysed against pH 7.5, the zeta potential was negative for all measured pH values, indicative of free miRNA. (**d**) In contrast, HiPerFect transfection reagent is positively charged at all measured pH value and (**e**) changed to negative upon complexation with miRNA. (**B**) Agarose gel electrophoresis analysis of PEG-LNP miRNA complexation capacity. PEG-LNPs/miRNA before dialysis (lane 2) and after dialysis against PBS, pH 7.5 (lane 3) or pH 5.5 (lane 4). Triton–X–100–treated PEG-LNPs/miRNA before (lane 5) and after dialysis against PBS, pH 7.5 (lane 6) or pH 5.5 (lane 7). Nuclease-treated PEG-LNPs/miRNA after dialysis against PBS, pH 7.5 or pH 5.5 (lanes 8 and 9, respectively). Free miRNA control (lane 10), which was completely digested after treatment with nuclease (lane 11). Marker (lane 1).

**Figure 2 pharmaceutics-15-01983-f002:**
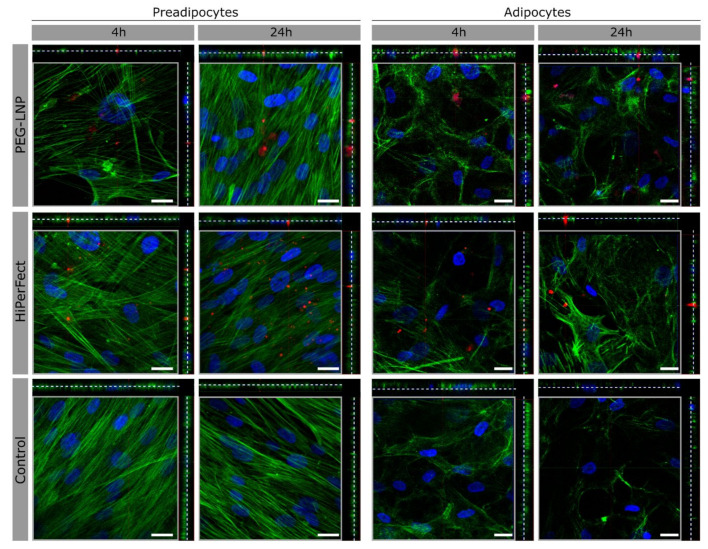
Confocal fluorescence images of preadipocytes (d-1) and adipocytes (d8) transfected with miRNA-CY3-complexed PEG-LNPs (**upper panel**) or HiPerFect (**middle panel**) as positive control for different time points (4 h and 24 h; non-transfected cells are shown as control (**lower panel**)). Nuclei were counterstained with Hoechst (blue). Alexa Fluor 488-Phalloidin was used for cytoskeleton staining (green). MiRNA-CY3 is seen in red color. Single vertical and orthogonal z-stack fluorescence images shown on the right and upper sides of each image, respectively, were obtained at depths ranging from 7 to 7.5 µm from the cell surface (marked by dashed lines). Scale bars represent 20 µm.

**Figure 3 pharmaceutics-15-01983-f003:**
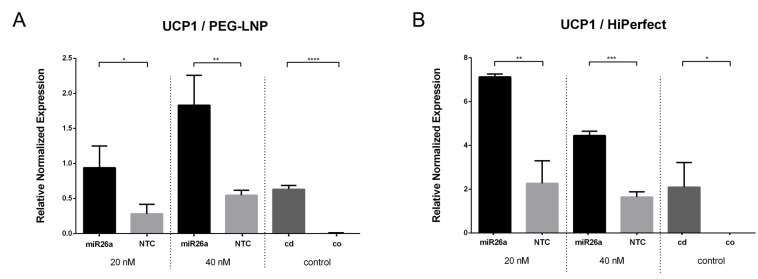
Expression profile of UCP1. RT-qPCR showing UCP1 target gene expression in SGBS adipocytes at d14-d16 (normalized to GAPDH, n = 3) after transfection treatment with miR26a (20 nM and 40 nM). Relative mRNA expression level of UCP1 of (**A**) PEG-LNP/miR26a (black) and PEG-LNP/NTC (grey) complexes as well as (**B**) HiPerFect/miR26a (black) and HiPerFect/NTC (grey) are shown. HiPerFect served as positive control. Non-transfected preadipocytes (cells only, co; light grey) and adipocytes (cells differentiated, cd; dark grey) served as negative controls. Student’s *t*-test, *p*-values are indicated. Data are shown as the mean ± SD and considered statistically significant when *p*  <  0.05 (* *p* < 0.05, ** *p*  <  0.01, *** *p*  <  0.001, and **** *p*  <  0.0001).

**Figure 4 pharmaceutics-15-01983-f004:**
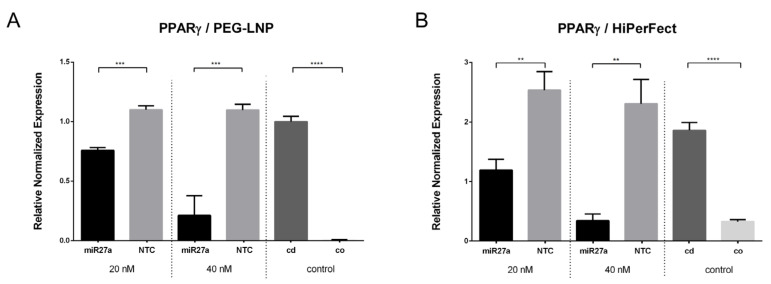
Expression profile of PPARγ. RT-qPCR showing PPARγ target gene expression in SGBS adipocytes at d5-d6 (normalized to GAPDH, relative to NTC, n = 3) after transfection with miR27a (20 nM and 40 nM). Relative mRNA expression of PPARγ of (**A**) PEG-LNP/miR27a (black) and PEG-LNP/NTC (grey) as well as (**B**) HiPerFect/miR27a (black) and HiPerFect/NTC (grey) are shown. HiPerFect served as positive control. Non-transfected preadipocytes (cells only, co; dark grey) and adipocytes (cells differentiated, cd; light grey) served as negative controls. Student’s *t*-test, *p*-values are indicated. Data are shown as the mean ± SD and considered statistically significant when *p*  <  0.05 (** *p*  <  0.01, *** *p*  <  0.001, and **** *p*  <  0.0001).

**Figure 5 pharmaceutics-15-01983-f005:**
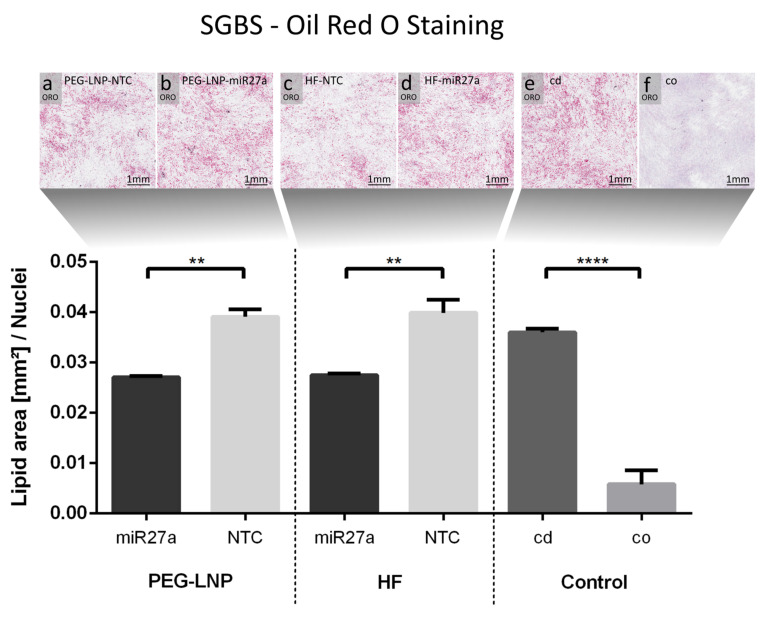
miR27a impedes lipid droplet formation during differentiation of SGBS preadipocytes to adipocytes. The cells were transfected at d1 and visualized by ORO staining six days after induction of differentiation as seen in the upper panel (**a**) PEG-LNP-miR27a, (**b**) PEG-LNP-NTC, (**c**) HiPerFect-miR27a, and (**d**) HiPerFect-NTC. Control conditions represent (**e**) adipocytes (cells differentiated, cd) and (**f**) non-transfected preadipocytes (cells only, co). Quantification of lipid-droplet-containing cells (lower panel) showed a significant decrease of the lipid area upon transfection with PEG-LNP-miR27a or HiPerFect-miR27a compared to PEG-LNP-NTC or HiPerFect-NTC, respectively. Differentiated adipocytes showed a similar high concentration of lipid droplets as cells transfected with NTC. Data are shown as the mean ± SD and considered statistically significant when *p*  <  0.05 (** *p*  <  0.01, and **** *p*  <  0.0001).

**Figure 6 pharmaceutics-15-01983-f006:**
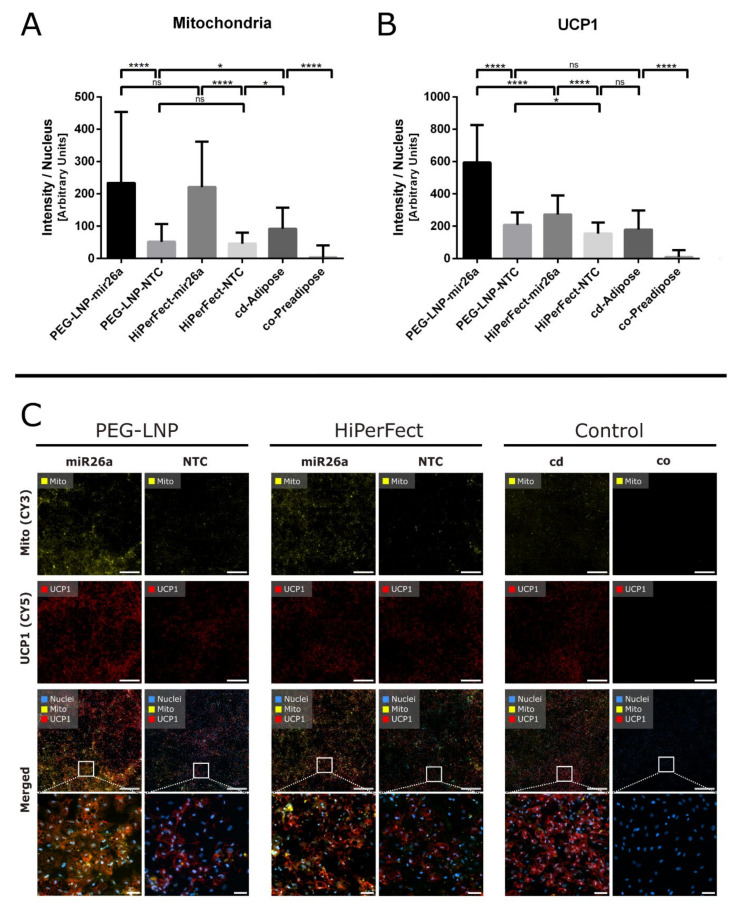
Quantitative image analysis of SGBS cells transfected with PEG-LNPs-miR26a at d8 show increased levels of (**A**) mitochondria and (**B**) UCP1 after immunostaining with MitoTracker™ Orange (mitochondria, CY3 channel) and UCP1 antibody (UCP1, CY5 channel) compared to transfection with NTC. HiPerFect-miR26a served as positive control. Untreated adipocytes (cd) and preadipocytes (co) served as negative controls. (**C**) Overview images of the evaluated fluorescence channels of over 12,000 cells. The first row of images shows the CY3 channel specific to fluorescence signals of mitochondria after different treatment conditions. The second row shows fluorescence signals specific to the UCP1 antibody (CY5 channel). The third row displays the combined channels for CY3, CY5, and DAPI (nuclei), while the fourth row displays a zoom-in of the merged images. The scale bars in the first three rows represent 1 mm, while the scale bars in the last row (zoom-in images) represent 100 µm. Data are shown as the mean ± SD and considered statistically significant when *p*  <  0.05 (* *p*  <  0.05, and **** *p*  <  0.0001).

**Table 1 pharmaceutics-15-01983-t001:** Summary of particle size data of miRNA-loaded and empty PEG-LNPs.

	Sample	Size [nm]	PDI
PEG-LNP miRNA	before dialysis	105.6 ± 2.6	0.233
	after dialysis pH 5.5	113.7 ± 1.9	0.246
	before dialysis	87.9 ± 2.3	0.102
	after dialysis pH 7.5	81.5 ± 1.8	0.094
PEG-LNP empty	before dialysis	109.6 ± 2.0	0.356
	after dialysis pH 5.5	111.4 ± 1.6	0.375
	before dialysis	87.9 ± 2.3	0.564
	after dialysis pH 7.5	93.4 ± 1.8	0.441

## Data Availability

Original data are available from the corresponding author upon request.

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
