# Peer review of "Lipid Nanoparticles as a Shuttle for Anti-Adipogenic miRNAs to Human Adipocytes"

_pharmaceutics, 2023, doi:10.3390/pharmaceutics15071983_

Round 1

Reviewer 1 Report

Introduction: The authors developed a pegylated lipid nanoparticle formulation containing two anti-adipogenic miR-23 NAs (miR26a and miR27a) for the potential treatment of obesity and related metabolic diseases such as type 2 diabetes. The developed nanoformulation were characterized for particle size, zeta potential, miRNA complexation efficiency and cytotoxicity. Overall, the manuscript is  well compiled.

Please make the following minor changes:

1.     Section 2.4, line 201 – Write SGBS in full before abbreviating in subsequent sentences since its at the beginning of the paragraph.

2.     Section 2.6, line 251 – Fix “Error! Reference source not found..” inserted in this section.

3.     Section 2.11, line 371 – Write SGBS in full before abbreviating since its at the beginning of the paragraph.

4.     Section 2.11, Line 377 – trademark for Cell Titer 96 should be superscript.

5.     Section 3.1, line 389 – I would recommend PEG-LNPs is written in full since its at the beginning of the paragraph.

6.     Figure 3: Legends/labels are not clear. Please improve quality of text.

7.     Line 564: Insert space in between to and 0.9

8.     Line 567: Insert space between 0.5 and to and to and 0.9-fold

9.     Figure 4: Improve clarity of the labels and legends of the graph

10.  In addition to the discussion section, I would recommend the addition of a separate conclusion section to the manuscript

None

Author Response

Reviewer #1

The authors thank the reviewer for the very positive assessment of our article.

Section 2.4, line 201 – Write SGBS in full before abbreviating in subsequent sentences since its at the beginning of the paragraph.

Section 2.6, line 251 – Fix “Error! Reference source not found..” inserted in this section.

 Section 2.11, line 371 – Write SGBS in full before abbreviating since its at the beginning of the paragraph.

Section 2.11, Line 377 – trademark for Cell Titer 96 should be superscript.

Section 3.1, line 389 – I would recommend PEG-LNPs is written in full since its at the beginning of the paragraph.

Answer: Thank you for the comments. We did all the corrections as suggested.

Figure 3: Legends/labels are not clear. Please improve quality of text.

Answer: Figure 3 was modified in a way to include a more detailed description of the x-axis to make the Figure clearer. We have also modified the Figure legend.

Line 564: Insert space in between to and 0.9

Line 567: Insert space between 0.5 and to and to and 0.9-fold

Answer: Thank you for the comments. We did all the corrections as suggested.

Figure 4: Improve clarity of the labels and legends of the graph

Answer: Figure 3 was modified in a way to include a more detailed description of the x-axis to make the Figure clearer. We have also modified the Figure legend.

In addition to the discussion section, I would recommend the addition of a separate conclusion section to the manuscript

Answer: Thank you for this comment. We have already discussed this issue before, but we think that adding a separate conclusion would just be a repetition of what we have already said in the discussion. I hope this decision finds the approval of the reviewer.

Reviewer 2 Report

This study was well performed in using a benchmark formulation for siRNA delivery to encapsulate miRNA and showed proof-of-concept in vitro transfection of preadipocytes and adipocytes. The targeted biological effects were achieved in well-controlled experiments, and compared against a commercially available transfection reagent. These progresses are valuable to be reported in the journal Pharmaceutics. I recommend publication of this manuscript; however, some aspects may need improvements before publication. Please find the summary of my comments below:

Some of the major points are:

** The title of the manuscript seems to be too broad, and seems more like a title for a review article. I would recommend revision of this title to better reflect the specific application of this manuscript, but I’ll leave this at the authors’ discretion.

** The pI and pKa values described for DLin-MC3-DMA in line 428 are off. Please note that the pI = 5.93 and pKa = 5.77 cited in the current manuscript from reference 53 are values of mRNA LNPs using DLin-MC3-DMA. These values of formed nanoparticles are different from pI and pKa values of the DLin-MC3-DMA lipid alone, as complexation with nucleic acid shifts the overall protonation behavior. This lipid should hold a pKa of around 6.4 to 6.5, and the significance of this value can be found in references such as this one: 10.1002/ange.201203263.

** In the descriptions related to zeta potentials, it may not be fully justified to describe a near-zero value as positive or negative, such as on line 423, -2.2 and 0.7 mV would both be considered “neutral on average”. Because the LNPs would have a distribution of zeta potential, only relative comparison between groups and between pH conditions would be meaningful, while their absolute values near neutral would not be as valuable to define. The authors may refer to this publication for this: 10.1016/j.jconrel.2016.06.017.

** While the gel results in Fig. 1B can be appreciated, it is routine to use quantitative assay, with RiboGreen as the most commonly accepted form, to measure encapsulation efficiency. The results would provide valuable information regarding the encapsulation of miRNA in this benchmark LNP formulation.

Here are some of the minor points that would improve the discussions in this manuscript:

** I would suggest the authors to clearly define “adipogenesis” at the beginning of the Introduction section as some descriptions currently in the manuscript seem to cause confusion for readers that are not familiar with this subject of matter. I feel it may be necessary to clearly state if it refers to the generation of WAT or BAT. Likewise, the description on page 2 line 75-77 “Inhibition of both miR26a family members significantly reduced lipid accumulation and the expression of characteristic adipocyte markers” is unclear in which markers and which type of tissues are being discussed.

** On page 3 line 121, it is unclear what “however” is referring to, as the preparation method described in this manuscript is not different from conventional microfluidics method in preparing the LNPs (a single mixing step followed by dialysis).

** For the summary of LNP size before and after dialysis into different pH buffers (line 389 to 411), it is recommended to generate a table to clearly summarize the findings with consistent statistical reporting “such as Mean ± S.D.” It would make it easier to compare sizes between groups.

Author Response

Reviewer #2:

The authors acknowledge the very positive assessment and thank the reviewer for appreciating our article. We thank the reviewer for his/her helpful comments which enabled us to improve our manuscript.

** The title of the manuscript seems to be too broad, and seems more like a title for a review article. I would recommend revision of this title to better reflect the specific application of this manuscript, but I’ll leave this at the authors’ discretion.

Answer: We agree with the comment of the reviewer and have modified our title.

** The pI and pKa values described for DLin-MC3-DMA in line 428 are off. Please note that the pI = 5.93 and pKa = 5.77 cited in the current manuscript from reference 53 are values of mRNA LNPs using DLin-MC3-DMA. These values of formed nanoparticles are different from pI and pKa values of the DLin-MC3-DMA lipid alone, as complexation with nucleic acid shifts the overall protonation behavior. This lipid should hold a pKa of around 6.4 to 6.5, and the significance of this value can be found in references such as this one: 10.1002/ange.201203263.

Answer: Thank you for this valuable comment. We have now corrected the value to pKa 6.44 citing the reference of Jayaraman et al (2012) on siRNA LNPs.

** In the descriptions related to zeta potentials, it may not be fully justified to describe a near-zero value as positive or negative, such as on line 423, -2.2 and 0.7 mV would both be considered “neutral on average”. Because the LNPs would have a distribution of zeta potential, only relative comparison between groups and between pH conditions would be meaningful, while their absolute values near neutral would not be as valuable to define. The authors may refer to this publication for this: 10.1016/j.jconrel.2016.06.017.

Answer: Thank you for this valuable comment. As suggested, we did the relative comparison between the same groups at different pH values.

** While the gel results in Fig. 1B can be appreciated, it is routine to use quantitative assay, with RiboGreen as the most commonly accepted form, to measure encapsulation efficiency. The results would provide valuable information regarding the encapsulation of miRNA in this benchmark LNP formulation.

Answer: We have repeated the preparation of PEG-LNPs following our SOP and have measured the miRNA concentrations using the RiboGreen assay. Free miRNA was used as reference. The samples have been measured directly after microfluidics assembly in 50 mM citrate buffer pH 4.0 containing 20% of ethanol and after dialysis against 10mM PBS buffer solutions (150mM NaCl; pH 7.5 or pH 5.5). In all cases the measured fluorescence intensity was low being lowest before dialyses and highest for the PEG-LNPs dialysed against pH 7.5. Upon addition of Triton X100 (final concentration 0.5% Triton X100) the PEG-LNPs were disrupted and the miRNA was released. The same concentration of miRNA was used as reference.  The concentrations of miRNAs were similar for all Triton X100 treated samples including the miRNA references c= 351.6 ± 38.2 µg/ml (n=7). A direct comparison with the fluorescence signal intensities before treatment with Triton X 100, revealed that about 11% of miRNA are released upon dialysis against PBS pH 7.5 while only 4 % are released when the samples are dialysed against PBS pH 5.5.

The assay is now included in Materials/Methods, particle preparation. The results are presented in

Here are some of the minor points that would improve the discussions in this manuscript:

** I would suggest the authors to clearly define “adipogenesis” at the beginning of the Introduction section as some descriptions currently in the manuscript seem to cause confusion for readers that are not familiar with this subject of matter.  I feel it may be necessary to clearly state if it refers to the generation of WAT or BAT.

Answer: We have now included a sentence stating that during adipogenesis adipocyte precursor cells develop into mature adipocytes which accumulate to form adipose tissue.

Likewise, the description on page 2 line 75-77 “Inhibition of both miR26a family members significantly reduced lipid accumulation and the expression of characteristic adipocyte markers” is unclear in which markers and which type of tissues are being discussed.

Answer: The sentence has been rephrased: … expression of characteristic adipocyte marker in adipose tissues including bioactive adipokines like adiponectin or leptin …. Including a reference.

** On page 3 line 121, it is unclear what “however” is referring to, as the preparation method described in this manuscript is not different from conventional microfluidics method in preparing the LNPs (a single mixing step followed by dialysis).

Answer:  According to the manufacturing protocol provided for Onpattro, we believe that the batches of Onpattro are fabricated by bulk mixing followed by ultrafiltration and exchange of buffer and not by controlled continuous flow microfluidics. See: https://www.ema.europa.eu/en/documents/assessment-report/onpattro-epar-public-assessment-report_.pdf)

Page 30/188:

Preparation of active substance and lipid solutions

  1. Mixing of solutions to form lipid nanoparticles (LNP)
  2. Ultrafiltration, exchange of buffer and initial concentration
  3. Dilution to final concentration and bioburden filtration
  4. Sterile filtration and filling into vials

We now have rephrased the sentence.

** For the summary of LNP size before and after dialysis into different pH buffers (line 389 to 411), it is recommended to generate a table to clearly summarize the findings with consistent statistical reporting “such as Mean ± S.D.” It would make it easier to compare sizes between groups.

Answer: We have now added Table 1 as suggested by the reviewer to make the comparison of sizes between the groups easier.

Reviewer 3 Report

In this research article, the authors presented “Lipid nanoparticle-based drug delivery: A shuttle for anti-adipogenic miRNAs to human adipocytes”. From my point of view, the topic is fascinating. The manuscript is concise and well-written. However, it has some issues to address before its publication in Pharmaceutics.

Following are my suggestions:

1)      Authors should add the product number of the chemicals they have used.

2)      Authors may also clearly write the amount of each chemical used for LNP synthesis. It will help others to repeat the synthesis.

3)      Authors may also design a schematic to help readers understand the work quickly.

4)      There is a reference error in the qPCR section. The authors should correct it.

5)      Why is there a lower expression of UCP1 upon an increase in the miR26a from 20 nM to 40 nM in the case of HiPerFect?

Author Response

Reviewer #3:

The authors thank the reviewer for the very positive assessment of the article and we also thank the reviewer for his/her helpful comments which enabled us to improve our manuscript.

Following are my suggestions:

1)      Authors should add the product number of the chemicals they have used.

Answer: We have now added the product numbers of the chemicals in the Materials and Methods section

2)      Authors may also clearly write the amount of each chemical used for LNP synthesis. It will help others to repeat the synthesis.

Answer: We have now added the amount of each chemical used for the LNP synthesis. The details are provided in chapter 2.1.

3)      Authors may also design a schematic to help readers understand the work quickly.

Answer: We would like to refer to the graphical abstract. The scheme is provided in the graphical abstract and we hope that the workflow is clearly depicted and understandable for the reader.  

4)      There is a reference error in the qPCR section. The authors should correct it.

Answer: Corrected

5)      Why is there a lower expression of UCP1 upon an increase in the miR26a from 20 nM to 40 nM in the case of HiPerFect?

Answer: To be honest we don´t know it. We did three independent experiments with HiPerFect and the normalized expression levels of UCP1 were 7.13 ± 0.075 and 4.45 ± 0.113 for 20 nM and 40 nM miRNA, respectively, with p < 0.00001.   Since HiPerFect served only as positive control and was not in the focus of our interest we did not pursue it further.